# Viral Infection and Airway Epithelial Immunity in Asthma

**DOI:** 10.3390/ijms23179914

**Published:** 2022-08-31

**Authors:** So Ri Kim

**Affiliations:** Division of Respiratory Medicine and Allergy, Department of Internal Medicine, Medical School of Jeonbuk National University, 20 Geonji-ro, Deokjin-gu, Jeonju 54907, Korea; sori@jbnu.ac.kr; Tel.: +82-63-250-2475; Fax: +82-63-259-3212

**Keywords:** asthma, viral infection, acute exacerbation, airway epithelial cells, inflammasome

## Abstract

Viral respiratory tract infections are associated with asthma development and exacerbation in children and adults. In the course of immune responses to viruses, airway epithelial cells are the initial platform of innate immunity against viral invasion. Patients with severe asthma are more vulnerable than those with mild to moderate asthma to viral infections. Furthermore, in most cases, asthmatic patients tend to produce lower levels of antiviral cytokines than healthy subjects, such as interferons produced from immune effector cells and airway epithelial cells. The epithelial inflammasome appears to contribute to asthma exacerbation through overactivation, leading to self-damage, despite its naturally protective role against infectious pathogens. Given the mixed and complex immune responses in viral-infection-induced asthma exacerbation, this review examines the diverse roles of airway epithelial immunity and related potential therapeutic targets and discusses the mechanisms underlying the heterogeneous manifestations of asthma exacerbations.

## 1. Introduction

In general, people with asthma are particularly susceptible to viral respiratory infections, which are a major cause of asthma exacerbation. Extensive basic and clinical studies in recent decades have contributed important insights about the role of viruses in asthma development and exacerbation and mediating factors related to those pathophysiologic processes. In addition, the severe acute respiratory syndrome coronavirus 2 (SARS-CoV-2) pandemic has highlighted the need for more effective preventive and therapeutic approaches to viral infections in patients with asthma, which require a deeper understanding of the interactions between asthma and viral infections.

The respiratory viruses mainly involved in asthma inception and exacerbation include rhinovirus (RV), respiratory syncytial virus (RSV), influenza virus, parainfluenza virus, adenovirus, and coronavirus [1,2,3]. Considerable epidemiological evidence supports the associations among RV infection, exposure, and sensitization to allergens with asthma onset and exacerbation [4,5]. Viral infections can cause asthma exacerbation via multiple mechanisms [2,3]: increased serum IgE levels, epithelial damage or activation, decreased antiviral responses (including the production of interferon (IFN)), alteration of host immune responses, promotion of inflammation in the respiratory tract, and direct infection of the lower respiratory tract.

In addition to their structural barrier function against allergens, infectious agents, and inhaled particulates, airway epithelial cells respond to various host and environmental stimuli by participating in diverse immune and inflammatory processes. In asthma, alterations in the airway epithelium are known to play critical roles in viral-infection-induced exacerbations, although contradictory results have been reported depending on the viral strain, cell type, experimental system (i.e., animal models vs. human subjects, in vivo vs. in vitro), and various host factors.

This review addresses the role of respiratory viral infection in asthma development and exacerbation, with a focus on airway epithelial immunity, including its structural barrier functions such as its integrity and repair, and its innate immune barrier function, including type 2 inflammation and inflammasomes.

## 2. Role of Respiratory Viral Infection in Asthma Pathogenesis

### 2.1. Role of Viral Infection in the Development of Asthma

Although the hygiene hypothesis suggests that early childhood infections are protective against allergic diseases later in life, including asthma [6], respiratory viral infections associated with wheezing illness are known to contribute to the development of asthma. Viral-infection–related pathology is influenced by host factors such as age, previous infection or immunization, pre-existing respiratory or systemic disease, and immunosuppression or compromise [7]. Viral respiratory disease can be caused by a localized respiratory tract infection, such as RSV infant bronchiolitis, or can be part of a generalized systemic illness, such as measles [7]. Viral respiratory infections are a major cause of wheezing in infants and adult patients with asthma. In particular, RSV and RV are important causes of wheezing in early life, and wheezing illnesses with these viruses have been associated with increased asthma risk later in childhood. Each year, RSV is the leading contributor to hospitalization in children younger than 1 year of age, whereas RVs are the most frequently detected viruses in wheezing children older than 1 year and in children and adults with acute exacerbations of asthma [8,9]. Sigurs et al. explored the association between severe RSV bronchitis and the eventual development of asthma [10]. They assessed disease progression in infants hospitalized for RSV bronchitis until they were 13 years old. At 7.5 years of age, children with RSV bronchitis and a family history of asthma were found to have higher asthma morbidity than healthy controls with a family history of asthma, suggesting that severe RSV infection with a family history of asthma increased the risk of developing asthma. In addition, some cohort studies have demonstrated an association between RSV-infection–induced wheezing illnesses in early life and the subsequent expression of persistent wheezing and asthma when a child begins school; the odds ratio (OR) for an asthma diagnosis in these studies was 2.6 (95% CI 1.0–6.3) [11,12]. In one of these studies, RV infection also turned out to be an important factor in the development of asthma in 6-year-old children who had experienced a related wheezing illness at age 3, with an OR for asthma of 9.8 (95% CI 4.3–22.0) [11]. These findings suggest that RV-induced bronchiolitis could be more strongly associated with the risk of developing wheeze and childhood asthma, which is also supported by a very recent meta-analysis [13]. The systemic review included 38 studies in the meta-analysis that directly compared between virus differences in the magnitude of virus-recurrent wheeze and virus–childhood asthma outcome. The analysis of the overall impact of RSV bronchiolitis on the development of recurrent wheezing or asthma in comparison to RV bronchiolitis showed that the RV bronchiolitis group was more likely to develop recurrent wheezing (OR 4.11; 95% CI 2.24–7.56) and asthma (OR 2.72; 95% CI 1.48–4.99) than the RSV group. More interestingly, an RV-infection–induced wheezing illness in infancy had a greater correlation with childhood asthma development than aeroallergen sensitization in infancy. Kusel et al. reported that the risk of asthma nearly doubled in children sensitized to common aeroallergens and increased by four times if more than two respiratory viral infections with wheezing were recorded during childhood. When the effects of allergic sensitization and a respiratory viral infection were evaluated together, their combination produced an approximately nine-fold increase in the risk of asthma, implying not only that atopy and viral infection are independent risk factors for developing asthma, but also that their combined effect seems to be synergistic [1,12]. Allergic sensitization and inflammation, particularly type 2 immune responses to allergens, are known to impair antiviral responses; however, the question of which comes first, type 2 inflammation or respiratory viral infections such as RSV- or RV-induced wheezing illness remains unanswered.

RV infections are very frequent at all ages in the general population, which begs the question of why only some of those infected with RV are at risk of developing asthma. In recent decades, many researchers have tried to answer that question by examining host genetic factors, viral strains, and environmental exposures. In terms of host genetics, polymorphisms in several antiviral and innate immune genes, including *STAT4*, *JAK2*, *MX1*, *VDR*, *DDX58*, and *EIF2AK2*, were linked to susceptibility to respiratory viruses, infection severity, and virus-induced asthma exacerbations [14]. In particular, variants in the 17q21 locus, including *ORMDL3* and *GSDMB*, were reportedly associated with an increased risk of RV-infection-induced wheezing in early life [15]. In children with an RV-infection-induced wheezing illness in the first 3 years of life, these variants were also associated with an increased risk of subsequent asthma. However, RSV-infection–induced wheezing was not linked to 17q21 variants in the same cohorts. Some environmental exposures, such as pets and farm materials, reduced the risk of asthma associated with the 17q21 genotype in children [16,17]. Meanwhile, RV-A and RV-C are more likely to show stronger virulence than RV-B, and thus, they are more likely to cause wheezing illnesses and lower respiratory infections [18]. A genome-wide association study of asthmatic children defined an association between asthma and a functional polymorphism in cadherin-related family member 3 (*CDHR3*) [19]. Considering that CDHR3 is a receptor that enables the binding and replication of RV-C, the link between CDHR3 and asthma risk might be mediated by RV-C infection [20]. In terms of the viral genome, the RSV A2, line 19, and Long strains have been used in experiments to define how the viral genome influences host immune responses to infections. The RSV A2 and Long strains produced similar immune responses in mice: predominant IFN-γ production, no production of IL-13 or airway mucus, and no airway hyperresponsiveness. However, RSV line 19 infection induced the production of IL-13 and airway mucus, reduced the production of IFN-γ, and produced exaggerated airway hyperresponsiveness [21,22]. When these strains were sequenced, differences were found in five amino acids in the fusion protein. Subsequently, a reverse genetic approach indicated that the fusion protein genes of the RSV line 19 strain were responsible for the lung production of IL-13 and airway mucus and airway hyperresponsiveness [22]. Throughout life, people are exposed to a variety of environmental factors, including respiratory pathogens, allergens, physicochemical irritants, and microbes. The influences of these exposures on human health, specifically in the development of asthma, remain the focus of research to define underlying pathogenic mechanisms and find new preventive and therapeutic interventions. From this point of view, epigenetics could be a promising approach to explaining the development of asthma associated with viral infection. Recent findings have shown that RV-induced alterations in DNA methylation are involved in the development and persistence of asthma [23,24]. In addition, several viral infections have been reported to contribute to the pathogenesis of asthma through the epigenetic expression of various non-coding RNAs (miRNAs). RV-infected asthmatic alveolar macrophages showed decreased TLR7 expression levels due to miR-150, miR-152, and miR-375 [25]. Mucus secretion was increased by a reduction in miR-34b/c-5p in RSV-infected human bronchial epithelial cells [26]. In epithelial cells from severe asthmatic patients, miR-22 was dysregulated by influenza A (IAV) infection, which could be one of the possible mechanisms of IAV-induced airway remodeling [27]. The microbiome, which is considered an endogenous environmental factor, may also contribute to respiratory viral infection. Teo et al. reported that the nasopharyngeal microbiome composition affected the infection severity and pathogen spread to lower airways, as well as the risk for future asthma development. Among the genera of bacteria in the nasopharyngeal microbiome, Streptococcus was a strong predictor, and antibiotic usage could disrupt the colonization patterns [28]. In a subsequent study, they revealed that viral-infection-associated respiratory illnesses were accompanied by a shift in the nasopharyngeal microbiome toward dominance by a small range of pathogenic bacteria genera. In addition, in conjunction with early allergic sensitization, the dominating presence of Streptococcus, Haemophilus, and Moraxella in the microbiome profiles of upper airways was a significant risk factor for persistent wheezing illness in school-age children, while these bacteria genera were associated with a transient wheeze that resolved in non-sensitized children [29].

### 2.2. Pathophysiological Effects of Viral Infection on Asthma Exacerbations

Respiratory viruses infect not only asthma patients but also healthy people. However, it is known that the pathological effects of respiratory viral infection in asthma patients are much more serious than those in healthy people. The clinical manifestations of viral infectious diseases are the results of both direct damage caused by the virus itself and damage caused by the host immune response to the virus. In an asthmatic patient, exacerbation can occur because of the functional interaction between the pathogenic effects of the virus and asthmatic inflammation [7]. Asthma exacerbation is characterized by an increase in fatal asthmatic symptoms, worse response to therapeutic controllers such as inhaled corticosteroids (ICS), and increased airway remodeling, which together can cause decreased lung function [30,31]. Viral respiratory infections are detected in about 85% of asthma exacerbations. Because asthma exacerbations are a major cause of morbidity in asthma patients of all ages, significant research efforts have been devoted to understanding the interaction between viral infection and asthmatic inflammation, particularly exacerbation [1]. Actually, the detailed immunological mechanisms associated with asthma exacerbation are currently unclear, although major advances in the research have improved our understanding of many aspects of the interaction between respiratory viral infection and underlying allergic asthmatic inflammation. The effect that coronavirus disease 19 (COVID-19), which is caused by SARS-CoV-2, has on acute exacerbations of asthma appears to be complex [32]. Indeed, some studies have reported that, unlike other respiratory viruses, SARS-CoV-2 rarely induces asthma exacerbations during hospitalization for COVID-19, that COVID-19-related asthma exacerbations have been relatively rare during the outbreak, and that SARS-CoV-2 pneumonia does not induce severe asthma exacerbation [33,34,35]. A recently updated report demonstrated that COVID-19 could lead to the worsening of asthma symptoms and prolonged exacerbation in some asthma patients, but not in all of them, and that allergic asthma patients had a significantly lower asthma exacerbation rate than non-allergic asthma patients [36]. These findings seem to be completely opposite to the effect of previous respiratory viral infection on asthma exacerbation, implying that the relationship between SARS-CoV-2 infection and asthma is complex and possibly unique among the known pathologic effects of previous viral infection. Some interesting studies have provided mechanistic insights into these observations [37,38]. It is known that the host cell entry of SARS-CoV-2 depends on angiotensin-converting enzyme 2 (ACE2), and the cellular serine protease transmembrane protease serine 2 (TMPRSS2) is used by SARS-CoV-2 for S protein priming [39]. Type 2 inflammation, or type 2 cytokine IL-13, suppressed the expression of ACE2 and increased the expression of TMPRSS2 in airway epithelial cells [37,38]. In addition to the strong negative influence of T2 inflammation on ACE2 expression in the airway, Sajithi et al. revealed an equally strong positive influence of respiratory virus infections on ACE2 levels [38]. They suggest T2 inflammation and virus-induced IFN inflammation as the strongest determinants of ACE2 and TMPRSS2 expression in asthmatic airway epithelium, which could contribute to the complex manifestations and various severities of COVID-19 in patients with asthma. Therefore, more work and data on the interaction between SARS-CoV-2 and asthma are needed to fully understand these intriguing findings.

Among the respiratory viruses, RV is the major and most frequent determinant of asthma exacerbation [40,41,42], although until recently, RV infection was known to occur primarily in the upper respiratory tract. However, several experimental infection models have directly implicated RV in lower airway infections involved in the pathogenesis of asthma exacerbation [43,44,45,46,47]. The proposed mechanisms for RV-induced asthma exacerbation include the activation of the airway epithelium, which produces an innate immune response and antigen-specific Th2 pathways that combine with allergic inflammation to enhance the overall type 2 inflammatory response [42,48,49,50]. In addition, RV infection could induce asthma exacerbation through a non-Th2 immune response, increased airway hyperresponsiveness, mucus hypersecretion, airway remodeling, or respiratory failure [51].

Asthmatic patients have increased susceptibility to viral respiratory infections, partly because they have deficient and delayed innate antiviral immune responses [52]. Many asthmatic patients tend to produce lower-than-average levels of type I IFN (i.e., IFN-α and IFN-β) and other cytokines in plasmacytoid dendritic cells (pDCs) and epithelial cells during viral respiratory infections [53,54,55,56]. This impaired antiviral immunity means that viral infections are associated with more severe airway damage in patients with asthma than in patients without asthma. Conversely, viral infection can increase the sensitivity of asthmatic airways to other triggers, such as allergens [7]. In addition, asthma is usually associated with pulmonary and extrapulmonary comorbidities, and these comorbidities are more common in severe asthma patients than in patients with mild to moderate asthma or those in the general population [57,58]. Pulmonary comorbidities include allergic rhinitis, obstructive sleep apnea, chronic rhinosinusitis (CRS), nasal polyps, chronic obstructive pulmonary disease, and bronchiectasis [58,59]. In particular, the mean prevalence of bronchiectasis in asthma patients is 36.6%, and patients with severe asthma with bronchiectasis show a higher rate of infection [60,61]. Bronchiectasis is frequently considered to be a consequence of long-lasting, severe, uncontrolled asthma, while asthma could be overlapped in patients with bronchiectasis [62]. Considering the structural and functional changes of bronchiectasis, such as mucociliary defects and biofilm formation, bronchiectasis is one of the comorbidities associated with the recurrent infectious exacerbation of asthma. Moreover, a recent study reported that respiratory viruses contributed to about 25% of the acute exacerbation of bronchiectasis and that IAV and RV made up over 50% of the viruses [63].

Recent interesting studies have reported that the diverse and distinct airway microbiomes of asthmatic patients can also influence viral respiratory infection, which is linked to acute exacerbation. McCauley et al. demonstrated that RV infection was more likely to occur in asthmatic children with Streoptococcus-species-dominated nasal airway microbiomes and that nasal microbiomes dominated by Moraxella species were associated with increased exacerbation risk and eosinophil activation [64]. In addition, a recent study indicated that specific networks of upper airway microbes (those possessing Streptococcus, Haemophilus, Neisseria, Prevotella, and other genera or those lacking Staphylococcus) that interacted with host transcriptional responses significantly increased the risk of subsequent exacerbation and that this relationship was also strongly dependent on season [65].

Taken together, it appears that virus-induced asthma exacerbation is the final consequence of a complex interaction among a variety of pathogenic mechanisms in pre-existing asthmatic inflammation: epithelial disruption and dysfunction, impaired antiviral immunity, inflammatory mediator overproduction, the induction of inflammation, IgE dysregulation, airway remodeling, alterations in neural responses, airway microbiomes, and differences in asthma endotypes and phenotypes.

## 3. Alteration of Airway Epithelial Cells in Viral-Infection-Induced Asthma Exacerbation

The airway epithelium is a pseudostratified columnar structure that protects against a variety of external stimuli by means of structural integrity, mucociliary clearance, and innate immunological barriers. These multifaceted barriers act cooperatively to maintain epithelial homeostasis and provide a dynamic response to pathogens, allergens, and environmental exposures [66]. The fundamental role and alterations of airway epithelium in asthma exacerbation have been well-documented. In addition, the recently introduced “epithelial barrier hypothesis” proposed mechanisms for the development of many chronic noncommunicable diseases, including asthma, through inflammation and tissue damage in the mucosal surfaces (i.e., epithelial barrier) of an affected organ or distant organs [67], which suggests that the airway epithelium could be a crucial preventive and therapeutic target for asthma and its exacerbation.

An initial target of virus-induced asthma exacerbation is the epithelium of the conducting airways. Once respiratory viruses enter airway epithelial cells and replicate within them, invasion depends on interactions with specific receptors, such as intracellular adhesion molecule 1 (ICAM-1), low-density lipoprotein receptors, CDHR3, sialic acids, nucleolin, cell surface integrin, dipeptidyl peptidase 4, and ACE2 [7,68]. It is important to recognize that the airway epithelium of an asthmatic differs significantly from a normal epithelium in ways that make it more susceptible to viral infection. This enhanced susceptibility to viral infection is known to originate from the destruction of the epithelium; the loss of ciliated cells; goblet cell hyperplasia; the upregulation of growth factors, cytokines, and chemokines; and impaired antiviral responses, such as the production of type I IFN.

In asthmatic pathology, the direct structural interruption of the airway epithelium provoked by injured tight junctions (TJs) and enhanced epithelial apoptosis results in epithelial leakiness [69]. Such a defective epithelium allows the entrance of respiratory pathogens, such as viruses and allergens, into subepithelial and deeper tissues, leading to antigen capture and presentation by the DCs [67,69]. In fact, RV infection dissociated TJ proteins, such as zonula occludens 1, from the TJ complex, and the protein levels of claudin-1 and occludin were significantly lower than normal in asthmatic children [70]. Additionally, RVs can dysregulate epithelial barrier function and integrity in several ways, including the disruption of homeostatic and dynamic cytokine production and TJ complexes and the dysregulation of wound repair [70,71,72,73]. In RV infection, defective repair was reported in asthmatic airway epithelium, which delayed wound repair and inhibited the normal apoptotic process [71]. In addition, RV-infected airway epithelial cells showed increased release of basic fibroblast growth factor and matrix metalloproteinase (MMP), leading to fibroblastic repair rather than normal epithelial repair, including abnormal cell death and extracellular matrix deposition [74]. A previous study also reported that asthmatic patients treated in an emergency room for acute exacerbation exhibited a significant increase in sputum MMP-9 levels compared with stable asthma patients and healthy subjects [75]. Supporting these data, a recent study examining cellular transcriptome networks revealed that the sequential upregulation of SMAD3, epidermal growth factor, and extracellular matrix after viral infection caused acute exacerbation in asthmatic patients [76]. Furthermore, a recent study using primary airway epithelial cells from pediatric asthma patients demonstrated an aberrant wound migration pattern associated with decreased integrin α5β1 expression, which is regulated by the PI3K/Akt pathway, suggesting that RV infection could disrupt the PI3K/Akt pathway, particularly in children susceptible to asthma [77]. These few mechanisms associated with deficient or dysregulated wound repair of the epithelium likely leave the epithelium susceptible to further infection or damage from exogenous insults. Virus-provoked immune responses are likely to amplify the overall inflammatory loads in subepithelial tissues, and the resulting deep tissue inflammation further disrupts the epithelial barrier. This vicious cycle potentiates the dysregulated subepithelial immune responses, inflammation, and remodeling that are aggravating factors for asthma exacerbation (Figure 1).

In terms of an innate immunological barrier, epithelial cell activation and the release of epithelial cell cytokines, such as the alarmins, IL-25, IL-33, and thymic stromal lymphopoietin (TSLP), play a significant role in causing and exacerbating allergic diseases such as asthma [78]. The alarmins act on subepithelial DCs, mast cells, and innate lymphoid cells (ILCs) to recruit both innate and adaptive immune cells and initiate the release of Th2 cytokines [79,80,81,82,83]. Indeed, a transcriptome network analysis showed that the IL-33 gene was upregulated in patients with virus-induced exacerbations compared with those with nonviral exacerbations [76]. RV infection induced IL-33 and Th2 cytokine responses in the airways of asthma patients, with IL-33 levels correlating with IL-5 and IL-13 levels. Blockage of the IL-33 receptor abolished RV-induced Th2 cytokine production by human T cells and type 2 ILCs [84]. Le Goffic et al. reported that, in both in vitro and in vivo experimental systems, influenza infection resulted in the expression and release of IL-33 [85]. Kaiko and colleagues also showed that primary pneumovirus infection in mice induced the expression and release of IL-33 [86]. Another study demonstrated that IL-33 could aggravate airway hyperresponsiveness and asthmatic inflammation through an innate immune response without Th2 cell involvement [87]. Supporting those observations, a recent study using a murine model showed that IL-33 suppressed innate antiviral responses and adaptive Th1 responses in influenza-induced exacerbations, which enhanced asthmatic airway inflammation [88]. Interestingly, it has been suggested that the cellular immune response to IL-33 following RV infection differs between people with and without asthma. Peripheral blood mononuclear cells (PBMCs) costimulated with IL-33 and RV showed that, although IL-33 augmented RV-induced IL-5 and IL-13 production in PBMCs from asthma patients, it had no effect on those from healthy controls. Additionally, IL-33 promoted the ILC production of IL-13 in asthma patients, whereas it promoted the natural killer cell production of IFN-γ in control subjects [89]. IL-25, a well-known contributor to the pathogenesis of asthma, was studied regarding its role in RV-induced asthma exacerbation; the RV infection of airway epithelial cells from asthma patients resulted in significantly higher IL-25 mRNA and protein expression than was found in cells from healthy controls [90]. Recently, an analysis of an immune transcriptome of RV-infected asthmatic airway epithelial cells treated with an anti-IL-25 monoclonal antibody revealed increased type I and III IFN levels and reduced expression of type 2 immune genes and the IL-25 receptor. The blockage of IL-25 also increased type I and III IFN expressions by airway epithelial cells infected with coronavirus. Exogenous IL-25 treatment increased the viral load and suppressed innate immunity, suggesting that IL-25 directly inhibited the innate antiviral immunity in airway epithelial cells [91]. TSLP was more greatly expressed in airway epithelial cells isolated from asthmatic subjects after RSV infection than in those from healthy controls, and a role for TSLP was defined using an RSV-infected TSLP-knock out mouse model in driving RSV-induced Th2 cells and their associated pathology [92]. An in vivo study using a mouse model with allergic asthmatic inflammation showed that pulmonary TSLP was induced exclusively during exacerbations evoked by RV infection or poly I:C [93]. In addition, TSLP production was increased by RV infection in a primary culture of bronchial epithelial cells [94]. Epithelial cytokines known as alarmins resulted in the production of IL-4, IL-5, and IL-13, which play key roles in asthma pathogenesis. Given the reciprocal negative regulation between the IFN and Th2 pathways, the overproduction of airway-epithelial-cell-derived cytokines is an attractive target for controlling virus-induced asthma exacerbation. Several anti-Th2 biologics have already shown impressive efficacy in reducing the rate of asthma exacerbations, particularly an anti-TSLP monoclonal antibody that, remarkably, lowered the annualized rates of exacerbations in patients with severe asthma, independent of their baseline blood eosinophil counts [95].

In addition to alarmins, the responses of airway epithelial cells to viral infection encompass a wide range of pro-inflammatory cytokines and chemokines, including eotaxins, RANTES, IL-17, TNF-α, IL-6, IL-8, and IL-1β [44,46,72,96,97,98,99,100]. Subauste and colleagues [46] demonstrated that RV infection induced TNF-α, IL-6, and IL-8 release in human bronchial epithelial cells and that prior exposure to TNF-α increased susceptibility to RV infection, suggesting that the cytokines could potentiate future RV infections. Asthmatic patients with an RV-C-induced wheezing illness showed increased levels of IL-17 and IL-1β, as well as enhanced Th2 cytokine release, in their nasal cytokine profiles [101].

In terms of IFN production by airway epithelial cells, a considerable number of studies have demonstrated that the host IFN response to viral infection is deficient in asthma patients [54,102,103,104,105,106,107], although deficient RV-induced epithelial IFN production in asthma has not always been observed. When airway epithelial cells from asthmatic patients and healthy controls were infected with RV, the IFN-β mRNA expression and protein production in cells from asthmatic patients were impaired compared with those from controls, resulting in increased viral replication in the cells of asthmatic patients. Conversely, the administration of exogenous IFN-β reduced viral replication in these epithelial cells [54]. Airway epithelial cells from asthmatic patients had a significant deficiency in the induction of type III IFNs, such as IFN-λ1 and IFN-λ2/3, following RV infection [104,105]. This IFN deficiency was also seen in children with severe asthma whose airway epithelial cells exhibited a high viral load that negatively correlated with IFN-β and IFN-λ mRNA levels [107]. Additionally, Holt et al. showed that the type I and III IFN response capacity appeared strongly constrained at birth. The risks for severe lower respiratory infections during infancy and the subsequent development of persistent wheeze were associated with a reduced capacity to respond to virus-related stimuli through the activation of type I and III IFN genes [108]. A recent study demonstrated that lower levels of epithelial IFN-α and IFN-β expressions correlated with more severe respiratory symptoms following RV infection, including decreased lung function and worse airway hyperresponsiveness [106]. However, a study assessing the kinetics of innate antiviral gene expression revealed that the RV-induced innate immune responses, including IFN production, of airway epithelial cells from asthmatic patients were delayed rather than deficient [96].

Double-stranded RNA (dsRNA)-activated serine/threonine kinase R (PKR) is well-characterized as an essential component of the innate antiviral response (i.e., it is downstream of the type-I-IFN-dependent signaling pathway). A recent study demonstrated that airway epithelial cells in bronchial smooth muscle from patients with severe asthma had enhanced susceptibility to RV infection, and viral replication within these epithelial cells occurred through the inhibition of the PKR pathway, which was associated with the increased secretion of CCL20, a novel mechanism for viral-infection-induced asthma exacerbation [109]. Interestingly, my previous report revealed somewhat opposite data: the phosphorylation of PKR in primary-cultured airway epithelial cells was enhanced in a mouse model of steroid-resistant severe asthma treated with poly I:C compared with control mice that did not receive the poly I:C treatment [110], and this aggravated asthmatic manifestations, including the production of epithelial alarmins, IL-25, IL-33, TSLP, and Th2 cytokines (unpublished data). In that study, a PKR inhibitor attenuated the features of severe asthma exacerbation, suggesting its therapeutic potential. Of course, a direct comparison of the results of these two experimental systems is limited considering that stimulation by Poly I:C and RV infection are completely different situations. Specifically, Poly I:C induces a TLR3 response, whereas RV induces a broad spectrum of PRRs. Thus, the conflicting results may have been due to differences in immune stimulation and response.

In this respect, to define the role of the airway epithelium in viral asthma exacerbation, more clear, well-designed experiments or trials of host factors (such as phenotypes) and viral factors are needed to understand these processes.

## 4. Role of the Epithelial Inflammasome in Viral Immunity in Asthma

The host innate immune response is characterized by critical mechanisms for promptly detecting, binding, and eliminating invading pathogens, such as viruses. Airway epithelial cells express various pattern recognition receptors (PRRs) for the rapid recognition of pathogens and pathogen-associated molecular patterns. Several families of innate PRRs, including Toll-like receptors (TLRs), C-type lectin receptors (CLRs), retinoic-acid-inducible gene I (RIG-I)-like receptors (RLRs), and nucleotide-binding oligomerization-domain-like receptors (NLRs), work cooperatively for host immunity [31,111]. PRRs are divided into membrane-bound and cytoplasmic PRRs: the membrane-bound forms include TLRs and CLRs, and NLRs and RLRs reside in the cytoplasm. The host response to respiratory viral infection begins with the recognition of viral RNA that enters the cytoplasm by endosomal TLR3, TLR7/8, RIG-I, and melanoma-differentiation-associated gene 5 (MDA5) [111,112]. TLR3 and TLR7 sense viral dsRNA and single-stranded RNA (ssRNA), respectively, within the endosome. RIG-I recognizes cytosolic ssRNA or viral RNA containing 5′-triphosphate and, by interacting with mitochondrial antiviral signaling (MAVS), induces type I and III IFN responses through nuclear factor-κB (NF-κB) and the IFN regulatory factor.

Inflammasomes are multimeric protein complexes composed of a sensor protein, such as NLRs, RIG-I, or MDA5; an adaptor protein, such as apoptosis-associated speck-like protein containing a CARD (ASC); and cysteine protease caspase-1, which leads to the maturation and secretion of pro-inflammatory cytokine IL-1β [113,114,115]. Five major inflammasomes have been well-identified so far: NLR family pyrin domain containing 1 (NLRP1), NLR family CARD domain containing 4, RIG-I absent in melanoma 2 (AIM2), and NLRP3 [116]. RIG-I signaling is important for activating the inflammasome via the MAVS-CARD9-NF-κB signaling pathway. In addition, RIG-I can directly activate the inflammasome complex by binding the adaptor ASC [115,117]. Among several NLR family members, NLRP3 is one of the most important intracellular PRRs and senses a diverse series of exogenous and endogenous danger signals, including infectious pathogens and sterile environmental stimuli [118,119,120]. In addition, NLRP3 inflammasome activation is crucial in the pathogenesis of several pulmonary inflammatory diseases, including asthma. Previous studies have revealed the pathogenic role of the NLRP3 inflammasome in airway epithelial cells, as well as in inflammatory and immune cells, from asthmatic murine models [121,122,123,124,125].

In terms of respiratory viral infection, four main inflammasomes are known to be involved in innate antiviral immunity against RNA viruses—NLRP3 and RIG-I, and in some cases, MDA5 and AIM2 inflammasomes [115,126,127,128]. Activation of the RIG-I and NLRP3 inflammasomes has been demonstrated in macrophages and DCs after infection with some respiratory RNA viruses, including RV [122,129], IAV [126,129,130], SARS-CoV-1 [131,132], and most recently SARS-CoV-2 [133,134]. However, data on the activation of epithelial inflammasomes by these viruses in airways, particularly in human airways, and their involvement in the pathology of asthma remain very poor.

Allen et al. showed that, when human airway epithelial cells were infected with IAV, they expressed NLRP3 and secreted IL-1β. In that study, mice lacking NLRP3 exhibited reduced innate immune responses; however, most of the NLRP3 response was derived from immune cells, not airway epithelial cells [126]. In addition, an in vitro study revealed that RV could activate the NLRP3 inflammasome in airway epithelial cells and mediate IL-1β [135]. The prevalence of asthma is increased in patients with CRS, and this strong association between asthma and CRS has been widely noted [136]. Using human primary nasal epithelial cells from patients with CRS, including asthma patients and healthy controls, a recent study showed that RV-induced epithelial NLPR3 inflammasome activation could mediate IL-1β secretion, cell pyroptosis, and mucin production in airway epithelium through the DDX33/DDX58-NLRP3-caspase-1-GSDMD-IL-1β signaling axis, and these pathologic changes were relevant to virus-induced acute exacerbation [137]. Reovirus, a dsRNA virus infection, induced a greater activation of the NLRP3 inflammasome in airway epithelial cells from EphA2-knockout mice than that in wild-type mice, which resulted in the production of IL-1β. Although reovirus is not a major viral pathogen in patients with acute asthma exacerbation, the study also showed that EphA2 suppressed the asthmatic inflammatory response in an ovalbumin-induced asthma murine model, suggesting that EphA2 functioned as a negative regulator of inflammasome activation upon reovirus infection by targeting epithelial NLRP3 and, thereby, guarding against an excessive, self-destructive immune response [138]. In addition to the NLRP3 inflammasome, a very recent study evaluated the role of the RIG-I inflammasome in respiratory viral infection, including RV and SARS-CoV-2, in asthma [139]. That study used controlled experimental in vivo RV infection in healthy controls and patients with asthma, as well as in vitro models of house dust mite (HDM) exposure and RV/SARS-CoV-2 coinfection, in primary airway epithelial cells from both groups and found that RV infection in patients with asthma led to the overactivation of RIG-I inflammasomes, which diminished RIG-I accessibility for type I and III IFN responses in airway epithelial cells, leading to their functional impairment, prolonged viral clearance, and unresolved inflammation in vivo and in vitro. Interestingly, prior infection with RV restricted SARS-CoV-2 replication, but coinfection with RV and SARS-CoV-2 augmented RIG-I inflammasome activation and epithelial inflammation in patients with asthma, especially in the presence of HDMs.

Given that the activation of inflammasomes is critical for protecting the host from invading viruses, such as RV, RSV, and IAV [140,141], the question arises as to which factors could overactivate the inflammasome, leading to unwanted tissue damage and the severe exacerbation of diseases such as asthma. To date, there is no definitive answer to this question. However, considering that the majority of research data on the role of the inflammasome is gathered from myeloid and immune cells, the cellular source of the inflammasome might be one of the keys. In particular, airway epithelial cells are composed of various cell types that each have a unique function; each cell could represent a different pattern of immunological response, and the role of the epithelial inflammasome could influence the fate of the overall inflammasome (protective vs. pathological) at the position in viral infections where the pathogens are initially recognized (Figure 2).

## 5. Current and Potential Therapeutics for Viral Infection in Asthma

Exacerbations occur across the spectrum of asthma severity and place a great burden on healthcare systems and patients [142]. Viral infections are among the most frequent causes of asthma exacerbation [1]. However, the prevention and treatment of viral-infection-induced asthma exacerbations remain unmet medical needs. Maintenance on ICS, the mainstay of asthma treatment, is effective in reducing the risk of asthma exacerbation, and the risk is further reduced by the use of inhaled, long-acting β-agonists. Indeed, guidelines emphasize the importance of using preventive therapies to reduce asthma exacerbation and suggest increasing the ICS dose when symptoms begin to indicate a loss in asthma control [143]. However, those interventions have shown mixed and equivocal results. A Cochrane Database review concluded that current evidence did not support increasing the ICS dose in patients with mild to moderate asthma as part of a self-management plan to treat exacerbation [144]. Moreover, given that exacerbations are more frequent in severe asthma with steroid resistance, simply increasing the ICS dose or using systemic steroids appears to be a limited approach to preventing and treating asthma exacerbations.

The new generation of anti-type 2 biologics has shown impressive efficacy in reducing the rate of acute asthma exacerbations [145,146,147,148,149]. They might also be efficacious during viral infections by improving the antiviral response. In a pediatric allergic asthma population, the use of omalizumab as a treatment adjunct for virus-induced exacerbations was shown to reduce the duration of RV infections, peak viral shedding, and the frequency of RV infections [150]. In another study, omalizumab significantly decreased the severity of RV-induced asthma exacerbations, even among patients who started with poor baseline disease activity [151]. IgE receptor activation increases host susceptibility to viral infection, and the use of omalizumab could help improve antiviral responses in asthmatics [152]. In fact, IgE receptor activation on pDCs from asthmatics reduced type I IFN secretion in response to IAV [153] and the type I and III IFN releases in response to RV compared with pDCs from nonasthmatics [55]. Furthermore, IgE cross-linking on PBMCs exposed to RV from asthmatics treated with omalizumab presented increased IFN-α secretion compared with a placebo group [145]. The efficacy of omalizumab was confirmed in a further prospective, observational cohort study of patients with allergic asthma from whom RV-positive nasal samples were collected. In that study, treatment with omalizumab resulted in a greater reduction in the severity of RV-induced exacerbations than treatment with ICS, even though the patients in the omalizumab group had greater disease activity than the ICS group at baseline [151]. Dupilumab might also improve antiviral immune responses in patients with T2-high asthma because IL-4 and IL-13 impaired viral-induced IFN production and TLR3 expression [154]. Biologicals targeting the IL-5 pathways—mepolizumab, reslizumab, and benralizumab—significantly decrease eosinophilic inflammation and the frequency of exacerbations, suggesting a potential role of eosinophils in virus-induced exacerbations. However, in a recent experimental RV challenge study in patients with mild asthma, mepolizumab showed no significant effect on decreased lung function or loss in asthma control after viral infection. Instead, the mepolizumab-treated group had higher viral loads in nasal swabs than the placebo group, suggesting that the type 2 immune response had a protective role against viral infection [155]. To define the precise role of the type 2 immune response in viral infection, further basic and clinical research is needed; however, to date, no specific clinical reports have indicated that anti-type 2 biologics could be harmful to asthma patients with viral infections. Moreover, several guidelines have recommended continuing biologic therapy in patients with severe asthma during the COVID-19 pandemic [143]. Although a recent report demonstrated that patients with severe asthma using biologic therapy were shown to have a more severe course of COVID-19 compared to the general population, it was not clear why the patients progressed to more severe COVID-19 because of the small number of cases analyzed. Out of a total of 634 severe asthmatic patients using biologics, only 9 patients contracted COVID-19 [156]. In addition, the incidence of COVID-19 and the risk of having COVID-19 were higher in severe asthmatic patients who interrupted their biological treatments, while the patients who continued biologic therapy showed good asthma control during the COVID-19 pandemic [157].

Given the evidence for IFN deficiency in the pathogenesis of virus-induced asthma exacerbations, the restoration of IFN seems to be a potential preventive and therapeutic approach to controlling viral infections in patients with asthma. A randomized, controlled trial with a 14-day regimen of inhaled IFN-β therapy following the onset of cold symptoms in asthma patients suggested that inhaled IFN-β could be beneficial in controlling virus-induced asthma exacerbations in severe asthmatics, although the trial did not meet its primary endpoint [158]. Fortunately, a recent phase-2 pilot trial was performed to assess the efficacy and safety of inhaled, nebulized IFN- β for the treatment of patients admitted to hospital with COVID-19 [159]. The treatment with inhaled, nebulized IFN- β seemed to be well-tolerated in patients with COVID-19, with a range of clinical outcomes displaying a beneficial pattern of response to the treatment. TLR3 monoclonal antibody, which targets PRRs as a TLR antagonist, was evaluated in patients with asthma. However, it was found to be ineffective at protecting against symptoms or decreases in lung function following RV infection, and it was associated with a greater number of moderate and severe exacerbations than the placebo group [160]. As a direct antiviral agent, palivizumab, a monoclonal antibody against RSV fusion protein, was reported to reduce subsequent recurrent wheezing in premature infants [161]. In addition, because ICAM-1 serves as a receptor for RV, an ICAM-1 blocker was studied for its ability to reduce RV-infection-induced acute asthma exacerbations. In humans, tremacamra, a recombinant soluble ICAM-1, reduced the severity of RV infection symptoms compared with the placebo group [162], but the high frequency of dosing it required prevented further clinical development. Several drugs targeting the RV capsid have been also studied. In patients with asthma, vapendavir had an antiviral effect, but it did not improve lung function or reduce exacerbation during RV infection [163]. To date, there is no official recommendation for the use of antiviral agents to reduce or prevent asthma exacerbations.

Data are lacking to show how effective respiratory virus vaccination is in reducing the rate of viral exacerbations in patients with asthma. In terms of influenza vaccination, a systematic review and meta-analysis demonstrated that the vaccine was safe and effective in asthmatics and could reduce the risk of asthma exacerbation, healthcare use, respiratory illness, and medications for asthma [164], but Cochrane data showed that the influenza vaccine did not reduce influenza-induced asthma exacerbations [165]. Nonetheless, an annual influenza vaccination is recommended for all individuals who do not have contraindications, especially persons at a high risk of infection, such as patients with asthma [166]. To date, no vaccinations against RV have been approved for use in humans. At present, COVID-19 vaccination is internationally recommended for asthmatics [143,167]. It is also advised that patients who are on biologics for asthma receive a COVID-19 vaccine. Recent reports have revealed that severe asthma patients on biologic treatment showed optimal safety and tolerability profiles of mRNA SARS-CoV-2/COVID-19 vaccines and that the biological treatment did not compromise the effectiveness and durability of the COVID-19-vaccine-induced immunity [168,169]. However, the long-term effects of COVID-19 vaccines on asthmatics are still unclear and warrant further investigation.

Recently, several studies have demonstrated the efficacy of bacterial immunotherapy using oral or mucosal formulations of polybacterial lysates (i.e., MV130 and OM85) for the prevention of viral wheezing illnesses or SARS-CoV2 infection [170,171,172,173]. Interestingly, in a clinical trial in children with wheezing attacks, sublingual treatment with MV130 showed safety and clinical efficacy against recurrent wheezing attacks [170]. In addition, using PMBCs from infants at high risk for asthma development, OM85 treatment primarily modulated gene networks triggered during innate immune responses to bacterial pathogens that typically accompany viral pathogens during severe lower respiratory infection [171]. As for SARS-CoV2 infection, OM-85 inhibited SARS-CoV-2 epithelial cell infection in vitro by downregulating the SARS-CoV-2 receptor, ACE2 expression, and TMPRSS2 transcription [172]. The prophylactic intranasal administration of MV130 conferred heterologous protection against SARS-CoV-2 infection in susceptible K18-hACE2 mice and improved the immunogenicity of two different COVID-19 vaccine formulations targeting the SARS-CoV-2 spike (S) protein when inoculated either intramuscularly or intranasally in C57BL/6 mice [173].

As potential therapeutics that could obtain an antiviral effect by targeting the epithelial barrier, celecoxib, a cyclooxygenase-inhibiting, nonsteroidal, anti-inflammatory drug, and azithromycin, a macrolide-class antibiotic, have been suggested [77,174]. Macrolides are the antibiotics most extensively studied for their therapeutic potential for asthma due to their antimicrobial, immunomodulatory, and possibly antiviral activities [175]. Some clinical studies have suggested that macrolides such as azithromycin and telithromycin can reduce asthma exacerbations in adults [174,176,177,178]. Azithromycin was shown to augment RV-induced IFN production in human bronchial epithelial cells and decrease RV replication in vitro [179]. The development of novel macrolides is also under investigation for their potential effects on IFN responses in airway epithelial cells and antiviral activity in cells from patients with asthma [180]. Defective airway epithelial cell repair following insults such as viral infection has been associated with asthma exacerbation. Celecoxib stimulated the PI3K/Akt-integrin α5β1 axis and restored airway epithelial repair in cells from children with wheeze [77]. Despite the shortage in supporting data, it has been postulated that these two drugs are useful for treating asthma exacerbations through their ability to restore epithelial barrier function. In addition, a TSLP monoclonal antibody, which targets epithelial-cell-derived cytokines, tezepelumab, could help attenuate exacerbations of asthma [70], but more evidence for this is needed. These potential therapeutics for viral infection in asthma are summarized in Figure 3.

## 6. Conclusions

People with asthma are more susceptible to viral infections than the general population, and this susceptibility appears to worsen with the increasing severity of asthma. Many studies—albeit primarily studies related to RV infection and RSV infection—have reported that viral infection is closely associated with the onset, progression, and exacerbation of asthma. The airway epithelial barrier is specifically positioned and wired to respond to viral infections. Recently, the epithelial cell barrier has attracted attention not only for its role as a structural barrier but also for its immunological role in the pathogenesis of asthma and viral-infection-induced exacerbation. Because the epithelial cell barrier is composed of various types of cells and there are differences in the directions and extents of immune responses affected by various factors, the related research has shown conflicting results. However, based on the data so far, asthmatic epithelial cell immunity seems to react differently from normal epithelial cells under viral infectious conditions, which appears to play a role in aggravating the disease by inducing self-damage through an excessive immune response rather than the normal protective function (Figure 1). Therefore, novel preventive and therapeutic drugs that target the epithelial cell barrier could hold promise for normalizing the immune responses of these cells, and it is helpful to consider how they can be applied to epithelial cells locally. Further research on and understanding of airway epithelial cell immunity, including normal antiviral defenses, are needed to suggest a new paradigm for asthma management in an era of relentless viral epidemics.

## Figures and Tables

**Figure 1 ijms-23-09914-f001:**
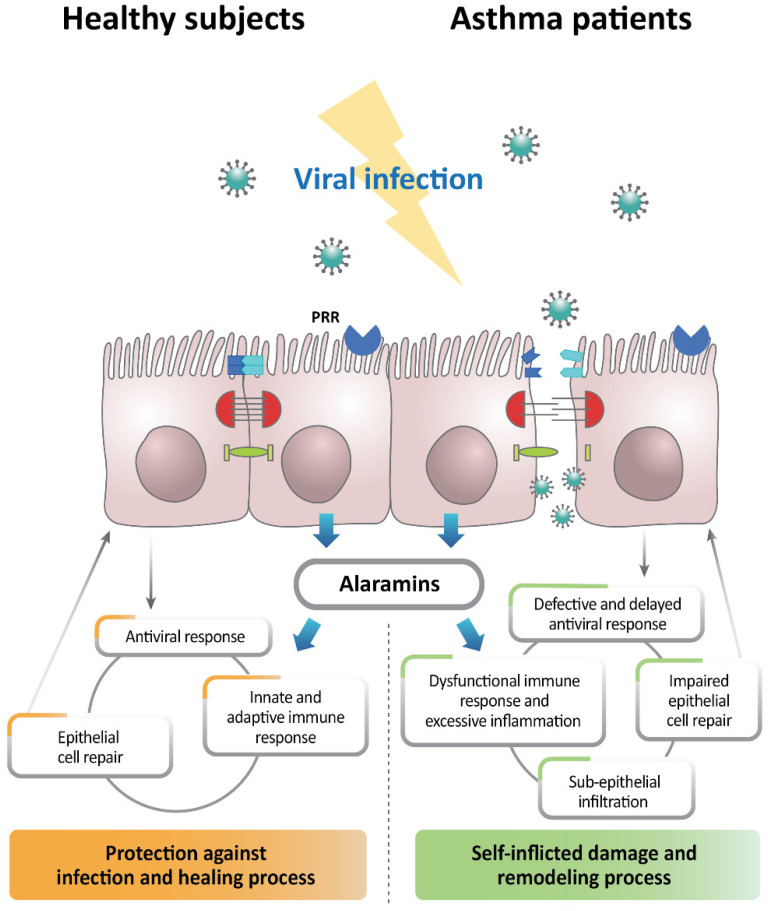
Overview of the airway epithelial barrier in asthmatics for viral infection compared to healthy airway epithelium. Asthmatic epithelial cell immunity reacts differently from normal epithelial cells under viral infectious conditions, which appears to play a role in aggravating the disease by inducing self-damage through an excessive immune response rather than the normal protective function. PRR; pattern recognition receptor.

**Figure 2 ijms-23-09914-f002:**
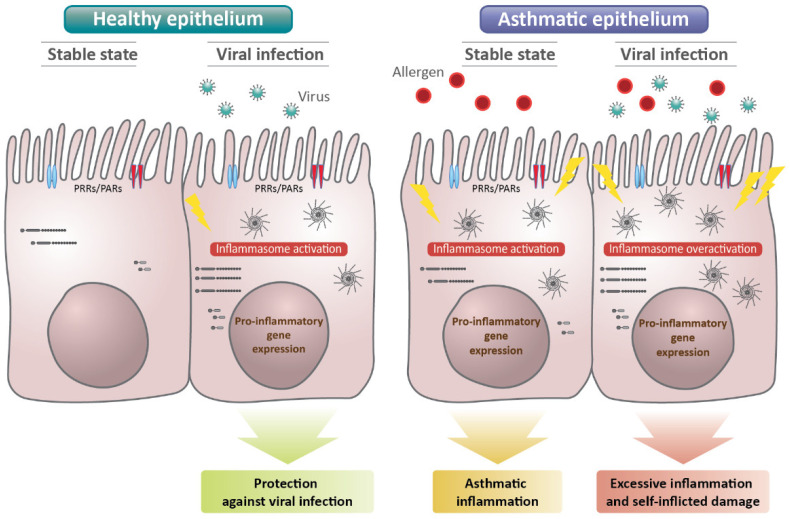
Proposed roles of the airway epithelial inflammasomes in asthmatics with viral infection compared to healthy airway epithelium. Asthmatic epithelial cells overactivate the inflammasome, leading to unwanted tissue damage and severe asthma exacerbation. PRR; pattern recognition receptor. PAR; protease-activated receptors.

**Figure 3 ijms-23-09914-f003:**
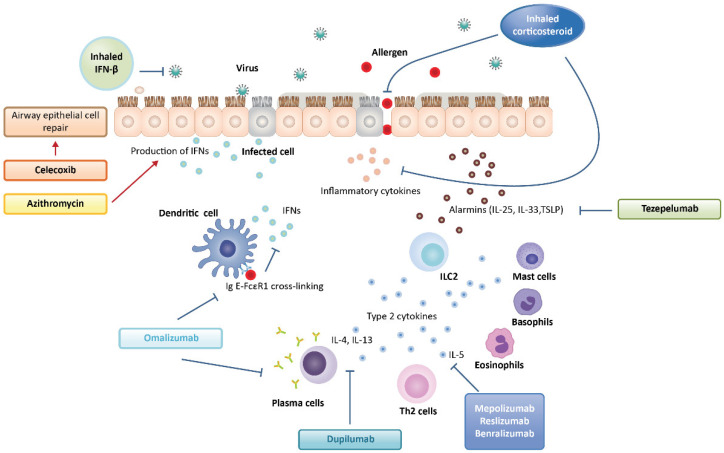
Schematic diagram of current and potential therapeutics for viral infection in asthma.

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
