# Peer review of "Viral Infection and Airway Epithelial Immunity in Asthma"

_ijms, 2022, doi:10.3390/ijms23179914_

Round 1

Reviewer 1 Report

Comments for the original version:

I have carefully read the Review entitled "Viral Infection and Airway Epithelial Immunity in Asthma".

The author's bibliographic research was very accurate and careful and it can be seen that the patient with severe asthma, in particular the eosinophilic Th2 phenotype, is particularly more vulnerable to viral infections.

However, some critical points emerge that need to be investigated in order to make the review more exhaustive.

1) In the phenotyping of severe Th2 eosinophilic asthma it is crucial to consider coexisting comorbidities and in particular also patients with bronchiectasis. It is now known in the literature that this phenotypic variant of severe asthma is quite widespread and often the treatment is influenced by the coexistence of bronchiiectasis which makes it more difficult to achieve and maintain control of asthma. Bronchiectasis besides predisposing to bacterial infections are a trigger for viral infections. Therefore it would be appropriate to include in a paragraph, even short, precisely on the comorbidities of asthma and susceptibility to viral infections in this context.

2) The use of biological drugs can influence the therapeutic response and the protective or non-protective response to viral infections. In fact, there are several works in the literature also on the role of biological therapies and SARS-CoV2 infection and also on the efficacy of vaccines.

Therefore a brief mention of the biological therapies used and the protective role or not that they can determine.

3) Improve Native English

4) Consider Splelling check.

Comments for the revised version:

I’ve read the paper with the proposed changes. For me the paper is acceptable without changes.

Author Response

REVIEWER 1

Comments for the original version:

General comments

I have carefully read the Review entitled "Viral Infection and Airway Epithelial Immunity in Asthma".

The author's bibliographic research was very accurate and careful and it can be seen that the patient with severe asthma, in particular the eosinophilic Th2 phenotype, is particularly more vulnerable to viral infections.

However, some critical points emerge that need to be investigated in order to make the review more exhaustive.

Major comments:

1) In the phenotyping of severe Th2 eosinophilic asthma it is crucial to consider coexisting comorbidities and in particular also patients with bronchiectasis. It is now known in the literature that this phenotypic variant of severe asthma is quite widespread and often the treatment is influenced by the coexistence of bronchiectasis which makes it more difficult to achieve and maintain control of asthma. Bronchiectasis besides predisposing to bacterial infections are a trigger for viral infections. Therefore it would be appropriate to include in a paragraph, even short, precisely on the comorbidities of asthma and susceptibility to viral infections in this context.

Response: I appreciate the reviewer’s insightful comment very much. I have agreed with the reviewer on this comment. I included the content on the comorbidities of asthma, in particular bronchiectasis, and susceptibility to viral infections in the revised manuscript [marked] (page 5, line 216-230).

2) The use of biological drugs can influence the therapeutic response and the protective or non-protective response to viral infections. In fact, there are several works in the literature also on the role of biological therapies and SARS-CoV2 infection and also on the efficacy of vaccines. Therefore a brief mention of the biological therapies used and the protective role or not that they can determine.

Response: I thank the reviewer for the comment. As the reviewer suggested, I added the content on the effects of biological therapies on viral infection including SARS-CoV2 and vaccination in asthmatics in the revised manuscript [marked] (page 12, line 551-559; page 13, line 594-599).

3) Improve Native English

Response: I thank the reviewer for the comment. As the reviewer commented, in order to improve native English, the manuscript has undergone professional English proofreading.

4) Consider spelling check.

Response: As the reviewer commented, spelling was re-checked throughout the manuscript.

Comments for the revised version:

General comments

I’ve read the paper with the proposed changes. For me the paper is acceptable without changes.

Response: I deeply appreciate the reviewer’s thoughtful and thorough review of our manuscript.

Reviewer 2 Report

Major comments

In this review article, the author provides an interesting and informative discussion of the role of respiratory viral infections, in particular RSV and RV, in the development of childhood asthma and as a trigger of acute exacerbations. The review primarily focuses on the central role of the airway epithelium, with respect to barrier function, innate immunity, and repair mechanisms. The discussion of innate immune responses focuses on the role of pathogen recognition receptors, alarmins, interferons, proinflammatory responses, inflammasomes, and ensuing Th2-mediated allergic inflammation. There is also discussion of novel therapies that have potential to modulate respiratory viral infections and subsequent asthma. Whilst the manuscript is interesting and informative, several important topics were overlooked or not covered in much detail. To address this issue, I have made some suggestions that would strengthen the review.

1. Role of viral infection in the development of asthma (Section 2.1)

In this section the role of RSV and RV in asthma development is described. Two studies were presented in which the odds ratios for asthma following RSV wheezing was 2.6 whereas the odds ratio for RV wheezing was 9.8. This section would benefit from a clearer statement to highlight that RV wheezing is a much stronger risk factor for asthma than RSV. The following meta-analysis can be cited to support this:

Makrinioti et al. Pediatr Allergy Immunol. 2022 Mar;33(3):e13741

2. The author discusses the finding that SARS-Cov-2 rarely induces asthma exacerbations but can lead to worsening of asthma symptoms and prolonged exacerbation (Line 154 – 168) but does not provide any mechanistic insights.

The following studies could be discussed to provide some insight.

Sajuthi et al. Nat Commun. 2020 Oct 12;11(1):5139

Kimura et al. J Allergy Clin Immunol. 2020 Jul;146(1):80-88.e8

3. Line 173: “Proposed mechanisms for RV-induced asthma exacerbation include the activation of the airway epithelium that produces an innate immune response and antigen-specific Th2 pathways that combine with allergic inflammation to enhance the overall type 2 inflammatory response [36, 42, 43]”

I would include an additional reference to support the argument around the role of antigen-specific Th2 pathways.

Muehling et al. J Allergy Clin Immunol. 2020 Sep;146(3):555-570

4. The Role of pathogenic bacteria in the pathogenesis of severe lower respiratory viral illness was not covered anywhere in the review.

Teo SM et al. Cell Host Microbe. 2015 May 13;17(5):704-15

Teo SM et al. Cell Host Microbe. 2018 Sep 12;24(3):341-352.e5

McCauley K et al. J Allergy Clin Immunol. 2019 Nov;144(5):1187-1197

McCauley K et al. J Allergy Clin Immunol. 2022 Jul;150(1):204-213

5. There is a discussion about the role of integrin a5B1 and PI3K signaling in dysregulated epithelial migration and repair. This discussion first appears on line 237 to 241 and is repeated again on lines 523 – 529.  

The manuscript should be edited to remove the repetitive statements.

6. The alarmins IL-25, IL-33, and TSLP are introduced on line 254 – 256. Later on, the author refers to epithelial cytokines known as pro-Th2 cytokines (line 294-295).

The epithelial pro-Th2 cytokines are not defined, and therefore I suspect that the author is referring to the alarmins, but this is not clear. Please clarify.

7. The following statement is misleading: “In mice, TSLP is more greatly expressed in airway epithelial cells isolated from asthmatic subjects than in those from healthy controls, which suggests a role for TSLP in driving RSV-induced Th2 cells and their associated pathology.”

How can TSLP be expressed in human asthmatic epithelial cells isolated from mice? Please clarify.

8. The author states that “In terms of IFN production by airway epithelial cells, a considerable number of studies have demonstrated that the host IFN response to viral infection is deficient in asthma patients [47, 86-91], although deficient RV-induced epithelial IFN production in asthma has not always been observed (Line 311 to 313).

The problem with these studies is that preexisting Th2 inflammation in asthmatic subjects inhibits interferon responses, and the intensity of Th2 inflammation is highly variable in asthma due to Th2hi and Th2lo asthma phenotypes. Therefore, it is not known if deficient production of interferons is a risk factor for disease development or alternatively is responding to Th2 inflammation in subjects with active disease. It is therefore important to consider the role of interferon responses in children before they develop asthma.

Holt et al. J Allergy Clin Immunol. 2019 Mar;143(3):1176-1182.

9. The author discusses contradictory findings from two studies with regards to the role of PKR signaling in airway epithelial cells in the pathogenesis of asthma (Line 329 to 342). Given that one study employed RV infection and the other study employed Poly-IC treatment, and that the findings were contradictory, it is important to acknowledge that RV and Poly-IC are not the same thing. Poly-IC will trigger TLR3 responses, whereas RV will trigger a broader range of pathogen recognition receptors (e.g. TLR2, TLR3, TLR7, TLR8, MDA5).

10. The following statement is misleading: “The host response to respiratory viral infection begins with the recognition of viral RNA and proteins that enter the cytoplasm by the endosomal TLR3, TLR7/8, RIG-I, and melanoma-differentiation-associated gene 5 (line 355 – 358).”

The pathogen recognition receptors describe here are relevant to viral nucleic acid not viral proteins. I would delete the reference to “and proteins” from this sentence.

11. The section on inflammasomes is not very clear or convincing and should be revised (line 392 – 414).

The first reference to the role of NLPR3 in RV responses is in patients with chronic rhinosinusitis. Accordingly, the relevance to asthma is not clear.

The next reference describes the role of NLPR3 in Reovirus infection in EphA2 KO mice. The relevance of Reovirus infection to asthma is not clear. Moreover, the relevance of EphA2 KO mice to asthma is not clear. It is important to better describe the relevance of these data to RV infection and asthma.

12. The following statement should be supported by references:

“Given that the activation of inflammasomes is critical for protecting the host from invading viruses such as RV, RSV, and IAV.” (Line 416)

13. The following statements are not clear and should be revised:

“the pathological contribution of inflammasomes raises some concerns, such as which factors over-activate the inflammasome, leading to unwanted tissue damage and severe exacerbation of diseases such as asthma (Figure 2). To date, there is no clear explanation for this issue (line 416-419).

14: The following statement is misleading: “A randomized controlled trial with a 14-day regimen of inhaled IFN-β therapy following the onset of cold symptoms in asthma patients suggests that inhaled IFN-β could be beneficial in controlling virus-induced asthma exacerbations in severe asthmatics.

It is important to mention that the clinical trial did not reach its primary endpoint and the therapy was abandoned for asthma. However, the drug was beneficial for patients with COVID-19.

Monk et al. Lancet Respir Med. 2021 Feb;9(2):196-206

15. The review did not mention the use of bacterial immunotherapy (OM85, MV130) for the prevention of viral wheezing or SARS-Cov2 infection.

Nieto et al. Am J Respir Crit Care Med. 2021 Aug 15;204(4):462-472

Troy et al. J Allergy Clin Immunol. 2022 Jul;150(1):93-103

Pivniouk et al. J Allergy Clin Immunol. 2022 Mar;149(3):923-933.e6

Del Fresno et al. Front Immunol. 2021 Nov 18;12:748103. doi: 10.3389/fimmu.2021.748103

Author Response

REVIEWER 2

General comments

In this review article, the author provides an interesting and informative discussion of the role of respiratory viral infections, in particular RSV and RV, in the development of childhood asthma and as a trigger of acute exacerbations. The review primarily focuses on the central role of the airway epithelium, with respect to barrier function, innate immunity, and repair mechanisms. The discussion of innate immune responses focuses on the role of pathogen recognition receptors, alarmins, interferons, pro-inflammatory responses, inflammasomes, and ensuing Th2-mediated allergic inflammation. There is also discussion of novel therapies that have potential to modulate respiratory viral infections and subsequent asthma. Whilst the manuscript is interesting and informative, several important topics were overlooked or not covered in much detail. To address this issue, I have made some suggestions that would strengthen the review.

Major comments:

1. Role of viral infection in the development of asthma (Section 2.1)

In this section the role of RSV and RV in asthma development is described. Two studies were presented in which the odds ratios for asthma following RSV wheezing was 2.6 whereas the odds ratio for RV wheezing was 9.8. This section would benefit from a clearer statement to highlight that RV wheezing is a much stronger risk factor for asthma than RSV. The following meta-analysis can be cited to support this: Makrinioti et al. Pediatr Allergy Immunol. 2022 Mar;33(3):e13741

Response: I appreciate the reviewer’s valuable comments very much. As the reviewer mentioned, I added the statement to highlight that RV wheezing is a much stronger risk factor for asthma than RSV and I cited the article suggested by the reviewer in the revised manuscript [marked] (page 2, line 79-87, ref. 13).

2. The author discusses the finding that SARS-Cov-2 rarely induces asthma exacerbations but can lead to worsening of asthma symptoms and prolonged exacerbation (Line 154 – 168) but does not provide any mechanistic insights. The following studies could be discussed to provide some insight: Sajuthi et al. Nat Commun. 2020 Oct 12;11(1):5139; Kimura et al. J Allergy Clin Immunol. 2020 Jul;146(1):80-88.e8

Response: I thank the reviewer for the insightful comment. As the reviewer commented, I added the detail content on the mechanistic insight of the finding that SARS-Cov-2 rarely induces asthma exacerbations but can lead to worsening of asthma symptoms and prolonged exacerbation and I cited the articles suggested by the reviewer in the revised manuscript [marked] (page 4, line 185-196, ref. 37, 38).

3. Line 173: “Proposed mechanisms for RV-induced asthma exacerbation include the activation of the airway epithelium that produces an innate immune response and antigen-specific Th2 pathways that combine with allergic inflammation to enhance the overall type 2 inflammatory response [36, 42, 43]”. I would include an additional reference to support the argument around the role of antigen-specific Th2 pathways: Muehling et al. J Allergy Clin Immunol. 2020 Sep;146(3):555-570

Response: As the reviewer commented, I included the article the reviewer suggested as an additional reference to support the argument around the role of antigen-specific Th2 pathways in the revised manuscript [marked] (ref. 50).

4. The Role of pathogenic bacteria in the pathogenesis of severe lower respiratory viral illness was not covered anywhere in the review: Teo SM et al. Cell Host Microbe. 2015 May 13;17(5):704-15; Teo SM et al. Cell Host Microbe. 2018 Sep 12;24(3):341-352.e5; McCauley K et al. J Allergy Clin Immunol. 2019 Nov;144(5):1187-1197; McCauley K et al. J Allergy Clin Immunol. 2022 Jul;150(1):204-213

Response: I thank the reviewer for the comments. As the reviewer commented, the role of pathogenic bacteria in the pathogenesis of severe lower respiratory viral illness was addressed with reference to the papers suggested by the reviewer in the revised manuscript [marked] (page 3, line 142-page 4, line 155; page 5, line 231-241  Ref.28, 29, 64, and 65).

5. There is a discussion about the role of integrin a5B1 and PI3K signaling in dysregulated epithelial migration and repair. This discussion first appears on line 237 to 241 and is repeated again on lines 523 – 529.  The manuscript should be edited to remove the repetitive statements.

Response: As the reviewer pointed out, the repetitive statements were edited in the revised manuscript [marked] (page 13, line 626-page 14, line 630).

6. The alarmins IL-25, IL-33, and TSLP are introduced on line 254 – 256. Later on, the author refers to epithelial cytokines known as pro-Th2 cytokines (line 294-295). The epithelial pro-Th2 cytokines are not defined, and therefore I suspect that the author is referring to the alarmins, but this is not clear. Please clarify.

Response: As the reviewer pointed out, the phrase “the epithelial pro-Th2 cytokine” was corrected to avoid confusion in the revised manuscript [marked] (page 8, line 350-351; page 8, line 359).

7. The following statement is misleading: “In mice, TSLP is more greatly expressed in airway epithelial cells isolated from asthmatic subjects than in those from healthy controls, which suggests a role for TSLP in driving RSV-induced Th2 cells and their associated pathology.” How can TSLP be expressed in human asthmatic epithelial cells isolated from mice? Please clarify.

Response: I thank the reviewer for the good point. As the reviewer pointed out, the statement was corrected to clarify in the revised manuscript [marked] (page 8, line 343-347).

8. The author states that “In terms of IFN production by airway epithelial cells, a considerable number of studies have demonstrated that the host IFN response to viral infection is deficient in asthma patients [47, 86-91], although deficient RV-induced epithelial IFN production in asthma has not always been observed (Line 311 to 313). The problem with these studies is that preexisting Th2 inflammation in asthmatic subjects inhibits interferon responses, and the intensity of Th2 inflammation is highly variable in asthma due to Th2hi and Th2lo asthma phenotypes. Therefore, it is not known if deficient production of interferons is a risk factor for disease development or alternatively is responding to Th2 inflammation in subjects with active disease. It is therefore important to consider the role of interferon responses in children before they develop asthma.: Holt et al. J Allergy Clin Immunol. 2019 Mar;143(3):1176-1182.

 Response: I appreciate the reviewer’s valuable comment. I agree with the reviewer on this issue. As the reviewer commented, consideration of the role of interferon responses in children before the onset of asthma was added to the revised manuscript [marked] (page 9, line 379-383, Ref. 108).

9. The author discusses contradictory findings from two studies with regards to the role of PKR signaling in airway epithelial cells in the pathogenesis of asthma (Line 329 to 342). Given that one study employed RV infection and the other study employed Poly-IC treatment, and that the findings were contradictory, it is important to acknowledge that RV and Poly-IC are not the same thing. Poly-IC will trigger TLR3 responses, whereas RV will trigger a broader range of pathogen recognition receptors (e.g. TLR2, TLR3, TLR7, TLR8, MDA5).

Response: I thank the reviewer for the very important point. I totally agree with the reviewer's comments. As the reviewer pointed out, the paragraph was corrected to acknowledge that RV and Poly-IC are not the same thing in the revised manuscript [marked] (page 9, line 403-407).

10. The following statement is misleading: “The host response to respiratory viral infection begins with the recognition of viral RNA and proteins that enter the cytoplasm by the endosomal TLR3, TLR7/8, RIG-I, and melanoma-differentiation-associated gene 5 (line 355 – 358).” The pathogen recognition receptors describe here are relevant to viral nucleic acid not viral proteins. I would delete the reference to “and proteins” from this sentence.

Response: As the reviewer pointed out, the words “and proteins” were deleted from the sentence in the revised manuscript [marked] (page 9, line 422).

11. The section on inflammasomes is not very clear or convincing and should be revised (line 392 – 414). The first reference to the role of NLPR3 in RV responses is in patients with chronic rhinosinusitis. Accordingly, the relevance to asthma is not clear. The next reference describes the role of NLPR3 in Reovirus infection in EphA2 KO mice. The relevance of Reovirus infection to asthma is not clear. Moreover, the relevance of EphA2 KO mice to asthma is not clear. It is important to better describe the relevance of these data to RV infection and asthma.

Response: I appreciate the reviewer’s valuable comment. I agree with the reviewer on the opinion, although the studies have reported the inflammasome’s role in respiratory epithelial cells. As the reviewer commented, to better describe the relevance of these data to RV infection and asthma, the manuscript was revised [marked] (page 10, line 457-460; page 10, line 464-468).

12. The following statement should be supported by references:

 “Given that the activation of inflammasomes is critical for protecting the host from invading viruses such as RV, RSV, and IAV.” (Line 416)

Response: As the reviewer pointed out, the references were cited in the revised manuscript [marked] (Ref. 140, 141).

13. The following statements are not clear and should be revised: “the pathological contribution of inflammasomes raises some concerns, such as which factors over-activate the inflammasome, leading to unwanted tissue damage and severe exacerbation of diseases such as asthma (Figure 2). To date, there is no clear explanation for this issue (line 416-419).

Response: I thank the reviewer for the good point. As the reviewer pointed out, to avoid confusion, the statement was corrected in the revised manuscript [marked] (page 10, line 485-page 11, line 489).

14. The following statement is misleading: “A randomized controlled trial with a 14-day regimen of inhaled IFN-β therapy following the onset of cold symptoms in asthma patients suggests that inhaled IFN-β could be beneficial in controlling virus-induced asthma exacerbations in severe asthmatics. It is important to mention that the clinical trial did not reach its primary endpoint and the therapy was abandoned for asthma. However, the drug was beneficial for patients with COVID-19.: Monk et al. Lancet Respir Med. 2021 Feb;9(2):196-206

Response: I thank the reviewer for the important comment. As the reviewer pointed out, it was mentioned that the clinical trial did not reach its primary endpoint and the therapy was abandoned for asthma. However, the drug was beneficial for patients with COVID-19 in the revised manuscript [marked] (page 12, line 565-570, Ref. 159).

15. The review did not mention the use of bacterial immunotherapy (OM85, MV130) for the prevention of viral wheezing or SARS-Cov2 infection.: Nieto et al. Am J Respir Crit Care Med. 2021 Aug 15;204(4):462-472; Troy et al. J Allergy Clin Immunol. 2022 Jul;150(1):93-103; Pivniouk et al. J Allergy Clin Immunol. 2022 Mar;149(3):923-933.e6; Del Fresno et al. Front Immunol. 2021 Nov 18;12:748103. doi: 10.3389/fimmu.2021.748103

Response: I appreciate the reviewer’s valuable comment. As the reviewer commented, the contention regarding the use of bacterial immunotherapy for the prevention of viral wheezing or SARS-CoV2 infection was incorporated into the revised manuscript [marked] (page 13, line 601-614, Ref. 170-173).

Round 2

Reviewer 2 Report

The author has addressed my comments. I have no further comments.